# Synergistic pretraining of parametrized quantum circuits via tensor networks

Manuel S. Rudolph[1], Jacob Miller [2], Danial Motlagh [1], Jing Chen[2], Atithi Acharya[2,3] & Alejandro Perdomo-Ortiz [1] ✉

Parametrized quantum circuits (PQCs) represent a promising framework for using present-day quantum hardware to solve diverse problems in materials science, quantum chemistry, and machine learning. We introduce a "synergistic" approach that addresses two prominent issues with these models: the prevalence of barren plateaus in PQC optimization landscapes, and the difficulty to outperform state-of-the-art classical algorithms. This framework first uses classical resources to compute a tensor network encoding a high-quality solution, and then converts this classical output into a PQC which can be further improved using quantum resources. We provide numerical evidence that this framework effectively mitigates barren plateaus in systems of up to 100 qubits using only moderate classical resources, with overall performance improving as more classical or quantum resources are employed. We believe our results highlight that classical simulation methods are not an obstacle to overcome in demonstrating practically useful quantum advantage, but rather can help quantum methods find their way.

In the coming years and decades, quantum computing resources will likely remain more expensive and less abundant than classical computing resources[1–3]. Despite the intrinsic theoretical advantages of quantum computers, the widespread adoption of quantum technologies will ultimately depend on the benefits they can offer for solving problems of high practical interest using these limited resources. To this end, parametrized quantum circuits (PQCs)[4–6] have been proposed as a promising formalism for leveraging near-term quantum devices for the solution of problems in quantum chemistry[7–9], materials science[10], and quantum machine learning[11–18] applications which are difficult for classical algorithms.

However, several distinct challenges stand in the way of reaching practical advantage over classical methods using parametrized quantum algorithms, such as the existence of *barren plateaus*[19–22] and unfavorable guarantees for local minima[23–25] in the PQC optimization landscape. Such results typically apply to the setting of generic PQC ansätze and parameter initialization schemes, and much less is known about scenarios where the initial parameters of a PQC are adapted for a particular task. While this adaptation has proven useful in quantum

chemistry, where circuits for computing molecular ground states have been shown to reach higher-precision results using initializations based on mean-field Hartree-Fock or more sophisticated coupled-cluster-based solutions (e.g., see refs. 26–29), task-specific initializations have seen much less use in other areas, such as *quantum machine learning* (QML).

Another difficulty for demonstrating an advantage over classical algorithms using PQCs is the increasing sophistication of classical simulation algorithms based on *tensor networks* (TNs), whose classically parametrized models can efficiently describe PQCs whose intermediate states have limited entanglement. The ability of TNs to be deployed on powerful classical hardware accelerators, such as graphical and tensor processing units (GPUs and TPUs), raises the bar for quantum hardware to overcome. This situation has led to a zero-sum game perspective on improvements in quantum vs. classical technologies, where advances in one domain are frequently viewed as placing additional burdens on practitioners of the other to attain relative advantages (see ref. 30 for a representative example).

[1]Zapata Computing Canada Inc., 325 Front St W, Toronto, ON M5V 2Y1, Canada. [2]Zapata Computing Inc., 100 Federal Street, Boston, MA 02110, USA. [3]Rutgers University, 136 Frelinghuysen Rd, Piscataway, NJ 08854, USA. ✉e-mail: aperdomo@post.harvard.edu

Compared to previous works, our method is broadly similar to the proposal of ref. 31 to use classically trained TN models for initializing PQCs, which was predicted to yield benefits in performance and trainability within general machine learning tasks. Our findings can thus be seen as both a concrete realization of this general proposal, applicable to a diverse range of circuit architectures and learning tasks, as well as a robust experimental verification of the benefits anticipated there. Closer to our work is the pretraining method of ref. 32, where trained MPS with bond dimension $\chi = 2$ were exactly decomposed into a staircase of two-qubit gates, which were then used to initialize quantum circuits for machine learning tasks. While this method was shown to improve the performance and trainability of PQC models, the restriction to $\chi = 2$ MPS placed a limit on the extent of classical resources which could be used to improve quantum models. By contrast, our synergistic optimization framework can be scaled up to utilize arbitrarily large classical and quantum resources, a difference that our results suggest gives continued returns in practice.

While the method we develop utilizes the specific circuit decomposition procedure of ref. 33, any other scalable MPS to PQC decomposition can be used in its place, so long as the following criteria are met: (a) It must accept as input MPS of arbitrarily large bond dimensions, (b) It must output a circuit of any desired depth formed from one- and two-qubit gates, and (c) It must converge to the original MPS state vector at a reasonable rate as the circuit depth increases. All of these criteria must be satisfied for the method to deliver the benefits seen here, with criterion (a) needed to use increasing classical resources (ref. 32 is limited here), criterion (b) needed to use increasing quantum resources within real-world quantum computers (the methods of refs. 34–36 output single-layer circuits of $k$-qubit gates, with $k$ unbounded), and criterion (c) needed to avoid fidelity plateaus which hinder the conversion of high-quality MPS into high-quality PQC (ref. 37 exhibits such fidelity plateaus[33]). Besides ref. 33, also the decomposition algorithms in refs. 38,39 satisfy all of these criteria, and are therefore promising candidates to be employed within this synergistic optimization framework.

In this work, we propose a synergistic framework for boosting the performance and trainability of PQCs using a pre-optimized initialization strategy built on scalable TN algorithms, which leverages the complementary strengths of both technologies. As depicted in Fig. 1, this method uses TNs to first find a promising quantum state for the parametrized quantum algorithm at hand, then converts this TN state to the parameters of a PQC, where further optimization can be carried out on quantum hardware. We employ a circuit layer-efficient decomposition protocol[33] for matrix product states (MPS), whose high-fidelity conversion of MPS to various PQC architectures allows leveraging high-quality MPS solutions. The resulting quantum circuits can be extended with classically infeasible gates which enable better performance relative to the MPS, as well as purely quantum-optimized circuits. We empirically verify these performance improvements in various problems from generative modeling and Hamiltonian ground state search, finding that our method successfully converts deep quantum circuits from being practically untrainable to reliably converging to high-quality solutions. We further give evidence for the scalability of our synergistic framework by probing the gradient variances, i.e., the "barrenness", of PQCs with up to 100 qubits, finding gradient variances and magnitudes to remain stable with increasing number of qubits and circuit depth. By ensuring that PQCs are set up for success using the best solution available with today's abundant classical computing resources, we believe that our methods might finally unlock the true potential of parametrized quantum algorithms as effective methods for solving problems of deep practical interest.

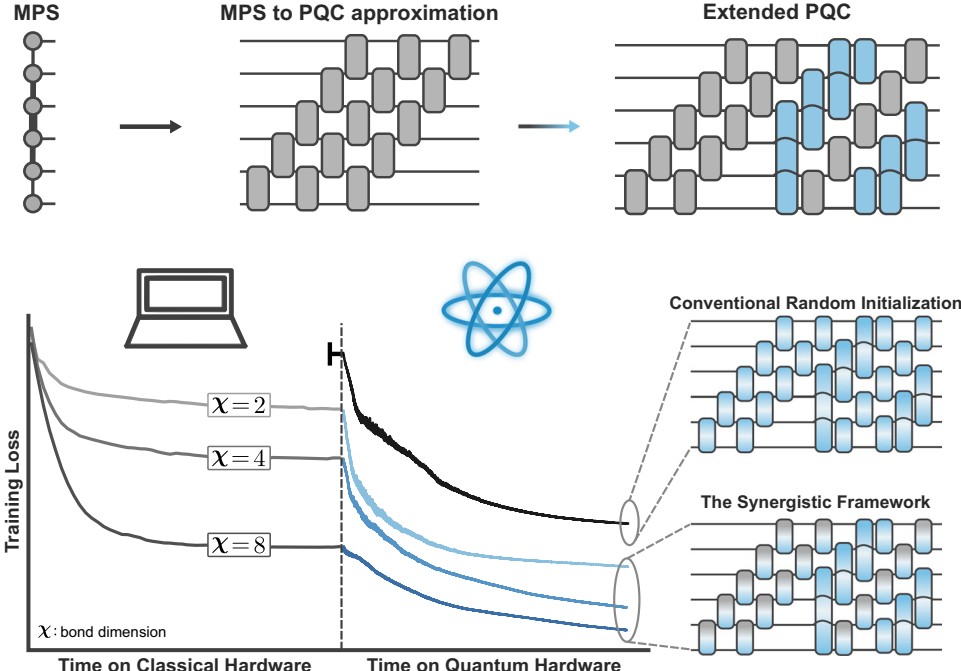

**Fig. 1 | Schematic depiction of the synergistic training framework utilizing TNs and PQCs.** Rather than starting with a random initialization of circuit parameters (black curve), which may suffer from problems such as barren plateaus and sub-optimal local minima, we instead train a matrix product state (MPS) model on a classical simulation of the problem at hand (left half of blue curves), whose performance is bounded by the limited entanglement available via its bond dimension $\chi$. This MPS wavefunction is then approximately transferred using a layer-efficient decomposition protocol that maps the MPS to linear layers of SU(4) gates. To improve on the classical solution, the quantum circuit is extended with additional gates (blue gates, initialized as near-identity operations) that would have been unfeasible to simulate on classical hardware. We find numerical evidence that quantum circuit models that leverage classically initialized circuit layers (gray & blue shaded gates) exhibit drastically improved performance over quantum circuits that were fully optimized on quantum hardware (blue shaded gates) and are likely to run into common trainability issues.

# Results

To assess the benefits of the synergistic training framework for real-world applications, we explore the performance of this method in a variety of generative modeling tasks, where the goal is to learn to reproduce the discrete measurement outcome distribution that is given by the training data, and a Hamiltonian ground state search problem. The corresponding MPS minimize the same loss function as the following PQCs, i.e., the KL divergence (4) for the generative modeling tasks, and the energy of the Hamiltonian for the ground state search. In each case, we compare quantum circuits trained using our MPS initialization approach to those which are initialized randomly or with all gates close to the identity. The latter has been shown empirically to reduce the effects of barren plateaus and improve convergence behavior at the start of PQC optimization[40].

We study the impact of different circuit architectures on our results by designing the quantum circuit layers with either linear or all-to-all topologies of fully parametrized SU(4) gates. While all-to-all connectivity is likely not practical in a scalable manner on near- to mid-term quantum hardware, it provides a challenging use case for an initialization method leveraging a TN model with linear connectivity, while also illustrating an important advantage that quantum hardware has over classical TN simulation techniques: The flexible choice of circuit depth and entangling topology. Implementation details can be found in Supplementary Note 4.

Our results find the use of TNs as a strategy to initialize the parameters of quantum circuits succeeds in boosting the performance of PQCs in all of these tasks, with an increase in classical computing resources (as quantified by the bond dimension $\chi$ of the MPS) in nearly every case leading to a corresponding increase in the final performance of the trained quantum circuit. This is reflected not only in the final losses in different tasks, but also through an analysis of the parameter gradients seen by the circuit at initialization. We find that although randomly initialized quantum circuits exhibit gradients of exponentially vanishing magnitude in system size, a manifestation of barren plateaus within generative modeling, the use of classically trained MPS to provide learned initialization avoids this phenomenon entirely.

In our first experiments, we explore the optimization performance of QCBM and VQE, i.e., the progression of the loss function values (Eqs. (4) & (5), respectively). We refer to Sec. IV A for details on these methods. For the QCBM and its TN equivalent, the TNBM (see Sec. IV B), we study two distinct datasets of bitstrings of length $N = 12$. The first is the *cardinality* dataset that is the set of all strings having a cardinality (i.e., Hamming weight, or number of 1s) of $N/2$. The second QCBM dataset is the dataset of bars and stripes (BAS) images[4,41] containing horizontal or vertical lines on a 2D pixel layout. The Cardinality dataset is a dataset with moderately low correlations between neighboring bits, whereas the BAS dataset is a dataset which exhibits strong correlations between bits within the same row or column, and thus makes it a 2d-correlated dataset. The VQE optimization problem uses $N = 9$ qubits and minimizes the energy of the 2D Heisenberg model Hamiltonian on a $3 \times 3$ rectangular lattice:

$$H = \frac{1}{4} \sum_{\langle i,j \rangle} \sigma_X^{(i)} \sigma_X^{(j)} + \sigma_Y^{(i)} \sigma_Y^{(j)} + \sigma_Z^{(i)} \sigma_Z^{(j)}. \qquad (1)$$

$\langle i, j \rangle$ indexes all nearest-neighbor spins in a 2D rectangular grid with open boundary conditions, and $\sigma_\mu^{(i)}$, $\mu = X, Y, Z$ denote the Pauli operators acting on the $i$'th spin. We measure the energy error $\Delta E(\boldsymbol{\theta}) = E(\boldsymbol{\theta}) - E_0$ relative to the exact ground state energy $E_0$

In all cases, we compare the performance of PQCs initialized with random SU(4) or near-identity unitaries to those initialized with previously found MPS solutions. Transferring the MPS state is done via the decomposition protocol described in ref. 33. As described in Sec. IV C, the topologies utilized here contain $k$ layers of gates, which are arranged linearly in the first $k-1$ layers and in an all-to-all topology in the last layer. For the cardinality dataset we utilize $k = 3$ layers, and for the BAS dataset, as well as for the 2D Heisenberg Hamiltonian, $k = 4$ layers. The parameters of the quantum circuits are optimized using the *CMA-ES* algorithm[42,43], a gradient-free optimizer that is based on an adaptive evolutionary strategy.

Our optimization results in Fig. 2 depict the best optimization runs out of 6 repetitions, i.e. the runs that reach the lowest loss after the prescribed training iterations. It becomes evident that the models without the MPS initialization do not converge to high-quality solutions. In fact, we have observed that, while all-to-all layers clearly increase the expressive capabilities of a PQC as compared to linear layers, the presence of a single all-to-all entangling layer has detrimental effect on their trainability (see also Supplementary Note 2). By choosing an initialization which makes use of the parameters of a classically trained MPS model however, all models exhibit a drastic increase in performance on all the tasks we considered. This behavior

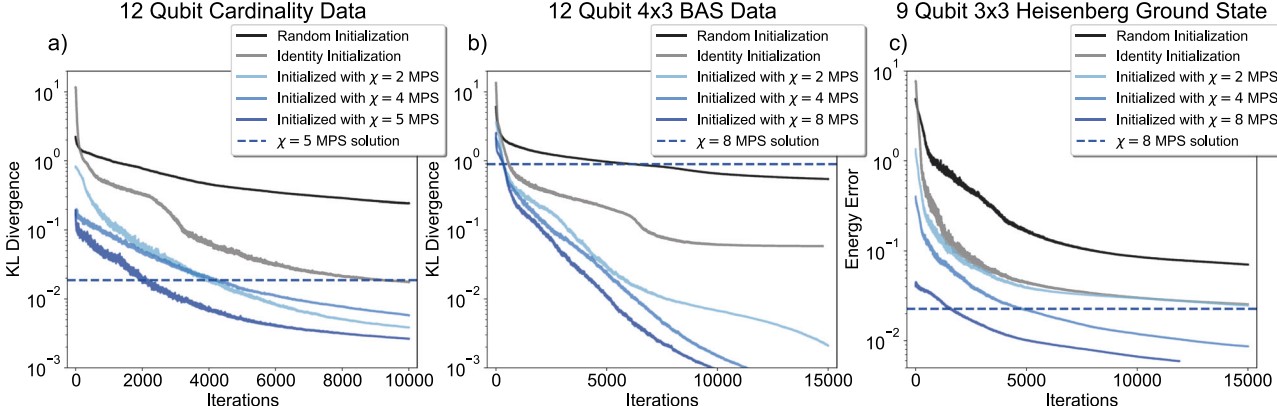

**Fig. 2 | Optimization results for QCBM and VQE training, where each curve represents the best performance among 6 independently initialized runs of the model.** The QCBMs are trained to minimize KL divergence relative to datasets of length $N = 12$ strings of fixed cardinality $N/2$ (**a**) or $4 \times 3$ bars and stripes images (**b**), whereas the VQE optimization (**c**) aims to minimize the energy of a 2D Heisenberg Hamiltonian with 9 qubits, and with size $3 \times 3$. The quantum circuits for each case consist of three, four, and four layers, respectively, with the SU(4) gates of each layer arranged in a linear topology, except for the final layer whose gates are connected in an all-to-all manner. In each case, quantum circuits whose parameters are initialized randomly or close to the identity exhibit worse final losses than those initialized with a classically trained MPS model. Additionally, the use of increased classical resources (as quantified by the bond dimension $\chi$) leads to improved performance of the trained quantum circuits. All optimization runs compared inside individual plots share exactly the same circuit layout and number of trainable parameters. Differences in training performance are only due to different initial parameters.

is enhanced even more by the use of MPS with larger bond dimension $\chi$. We emphasize that all PQC models compared inside one plot have precisely the same circuit layout and number of parameters. Differences in performance are only due to different choices of parameter initialization.

One may expect that the informed initialization merely affects the number of quantum circuit evaluations needed to achieve a target training loss, however, we show that it can also qualitatively change convergence behavior. For example, in the case of the cardinality dataset in Fig. 2a, QCBMs initialized with $\chi = 4$ or $\chi = 5$ TNBMs begin training at approximately the same KL values, but the PQC initialized with the larger $\chi$ TNBM rapidly achieves better loss values shortly after. Similar behavior can be seen in the case of the more challenging BAS dataset. In this case, the MPS solutions achieve relatively high KL divergence values, which consequently leads to high initial KL values for the MPS-initialized QCBMs. While the randomly initialized circuits generally reach the KLs at which the MPS initialized models start out, the latter are able to converge fully, while the former appear to plateau at much higher KL values.

Crucially, for the VQE optimization example in Fig. 2c we observe that initializing with $\chi = 2$ MPS does not suffice to reliably improve over a naive near-identity initialization of the circuit unitaries. Only MPS with larger bond dimension $\chi$, facilitated by the layer-efficient decomposition in ref. 33, enable significant enhancements. This is also highlighted by the depicted loss values achieved by the MPS solutions with highest $\chi$ that were used to initialize the respective PQC models. In the VQE simulation, initializing the PQC with $\chi = 2$ is not sufficient for the PQC to outperform what the MPS on classical hardware may have been capable of. The particular case of $\chi = 2$, where the MPS readily maps to two-qubit gates, was studied in ref. 32, and does not require the layer-efficient decomposition scheme used in this work which enables arbitrary $\chi$. However, the final MPS losses (indicated by the dashed lines) also showcase how the PQC solutions can improve on solutions attained on classical hardware by leveraging the more flexible capabilities of quantum hardware and initializing with strong classical models. The gaps between the final MPS losses and the respective PQC initializations stem from imperfect decomposition of the MPS into a low number of two-qubit gate layers, as well as the close-to-identity extension of the quantum circuits into the all-to-all topologies, and the initial exploration step size of the CMA-ES optimizer.

While initializing of the QCBM with a $\chi = 8$ TNBM on the BAS dataset in Fig. 2b here achieves the best result, we note that it does not clearly outperform $\chi = 4$ on average (see Supplementary Fig. 3). The likely reason is that the BAS dataset is 2D-correlated and thus the MPS with growing bond dimension $\chi$ increasingly biases the quantum circuit to a 1d-correlated solution. In other words, there is a bias mismatch between the TN architecture used and the task at hand. Depending on the number of additional free parameters that the PQC is given access to, this can lead to saddle points and local minima, because the PQC needs to correct the unsuitable bias. In such cases, one may try to train another TN model which is adapted for more general correlation structures[44], and then, if needed, map this TN to a quantum circuit[44,45]. Future work will need to study how to best extend pretrained quantum circuits with additional gates, i.e., where to most efficiently place additional gates such that the PQC can improve on the TN solution and potentially escape its bias.

To assess whether the synergistic framework is expected to be effective at improving the trainability of PQCs as the number of qubits increases, we now assess the variance of parameter gradients, i.e. the *barrenness*, of QCBMs training on the cardinality dataset. The results are shown in Fig. 3. We probe the gradient of the KL divergence loss with respect to the parameter controlling the YY-entangling component (according to the KAK-decomposition[46]) of the first SU(4) gate between qubits 1 and 2 (see Supplementary Note 4.B for details). Gradient magnitudes for that parameter are recorded 1000 times per data point in the case of random parameters, and 100 times per data point in the case of the TNBM initialization with $\chi = 2$. The latter case contains the training of the MPS, as well as the mapping to a quantum circuit, and the (potential) extension of the linear layers to all-to-all topologies. We note that our results are robust to different choices of the parameter for which the gradients are estimated.

In the case of randomly initialized parameters ($\boldsymbol{\theta} \in [0, 2\pi]$), we observe a clear exponential decay in gradient variances with increasing circuit depth and number of qubits. The nature of this decay depends on the quantum circuit topology used, with a single all-to-all layer being sufficient to saturate the barrenness for a specific number of qubits up until $N = 18 - 20$. In contrast, we observe that QCBMs initialized with a classically trained MPS avoid this exponential decay – something which likely contributes to the increased trainability observed in Fig. 2. Fascinatingly, the gradients in this initialization can

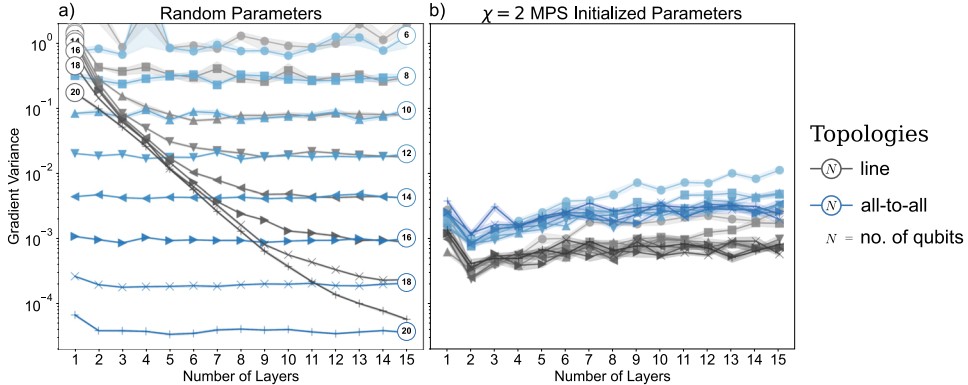

**Fig. 3 | Numerical evidence for the prevention of barren plateaus in QCBMs inside our synergistic optimization framework as demonstrated by the gradient variances with respect to the KL divergence loss in Eq. (4).** The gray lines indicate the gradient variances of linear topology circuits, whereas the blue lines indicate the gradient variances of all-to-all topology circuits. The numbers at the beginning or the end of the lines denote the number of qubits. We record the median gradient variances over 1000 repetitions for the randomly initialized circuits (**a**), and 100 repetitions for the MPS initialized circuits with bond dimension $\chi = 2$ (**b**), as well as bootstrapped 25-75 percentile confidence intervals of the median inside the shaded areas. We study the Cardinality $N/2$ dataset for the respective number of qubits $N$. The gradients are measured with respect to the YY-entangling gate contribution of the first SU(4) gate in the circuit between qubit 1 and 2. For random parameter initializations, the gradient variance decays exponentially in the number of qubits, and also the circuit depth up until a certain limit. This is clear indication for the existence of barren plateaus. One all-to-all layer of SU(4) gates appears to fully maximize the degree of barrenness. In contrast, the gradient variances of MPS-initialized circuits neither decay significantly in the number of qubits nor with increasing circuit depth.

actually exhibit an increase in variance with circuit depth, a trend which is more visible in the all-to-all extended circuit. Overall, the gradients of the all-to-all topologies have a larger gradient variance, indicating that the circuit extension after transferring the MPS solution is crucial to the success of the PQC. These findings suggest that the use of trained MPS places the quantum circuit model in a region of the parameter space without evident barren plateaus, but where the additional flexibility provided by increased connectivity in the quantum circuit enables it to effectively improve on the classical MPS solution. With a more sparse set of measurements, we identify a very similar trend when utilizing $\chi = 4$ MPS solutions.

Several potential criticisms may be raised about the scenario studied above and presented in Fig. 3. First, while statevector simulation allows us to generate valuable statistics for deep circuits, it only permits us to consider system sizes and datasets up to 20 qubits. This limitation is particularly restrictive when attempting to highlight the scalability of our method since trainability issues induced by barren plateaus are expected to manifest themselves more prominently as the qubit count increases. Consequently, we had to extend the decomposed circuits into an all-to-all topology to showcase the utility of our method more discernibly at such a limited qubit count. This is a second potential criticism because the study of all-to-all topologies is unlikely to be highly relevant in practice given the sheer number of noisy gates and possibly restricted hardware connectivity. Finally, the correlation structure in the Cardinality dataset is such that an MPS with bond dimension $\chi$ linear in the number of qubits can exactly represent the target distribution. Consequently, one might expect pretraining using an MPS to be abnormally successful. This fear is only partially supported by our findings in Fig. 2 because, while initial losses after pretraining on the BAS dataset are high, the resulting QCBM optimization is most dramatically improved.

We aim to address all these potential concerns with a complementary gradient scaling result using MPS-based quantum circuit simulation and a generative modeling task on the BAS dataset in a square arrangement. The 2D correlations in the BAS dataset suggest that a favorable circuit ansatz for a QCBM is one comprised of SU(4) gates in a 2D next-neighbor topology. Notably, this resembles a practical circuit topology for which quantum devices could exhibit an advantage, given the hardness of many 2D problems and the hardware connectivities in various modern quantum devices.

For the benchmarks, we train $N$-qubit TNBMs with $\chi = 2$ and $\chi = 4$ on all $\mathcal{O}(2^{\sqrt{n}})$ samples from the $\sqrt{n} \times \sqrt{n}$ BAS dataset. We then decompose the corresponding MPS into one linear layer of SU(4) gates and extend that layer into a 2D topology using identity-initialized SU(4) gates. For the random quantum circuit reference case, the linear part of the topology is randomly initialized, but the extension to the 2D topology is again done using identity operations. The gradients are computed via automatic differentiation of the MPS-based quantum circuit simulator. The identity initialization of the additional gates helps us simultaneously keep the gradient computations both feasible and exact by avoiding the need for the truncation of the simulator MPS.

Fig. 4 depicts the scaling of the gradient magnitude of the KL divergence loss function with respect to the circuit parameters, i.e., the 2-norm of the gradient vector, up to $10 \times 10 = 100$ qubits. Even in this new numerical setup, we observe results that are exactly consistent with the results in Fig. 3 for the $\chi = 2$ case, but we are now able to see that pretraining using a $\chi = 4$ MPS eventually outperforms and keeps up the favorable scaling. This supports the intuition that increasing classical resources are required as the problem size increases, and that high-performance schemes to convert tensor network states into quantum circuits will be needed in the future. However, it also suggests that moderate classical resources are sufficient in order to continue to provide value for the following quantum circuit optimization. One may have expected that drastic increases in classical compute would be

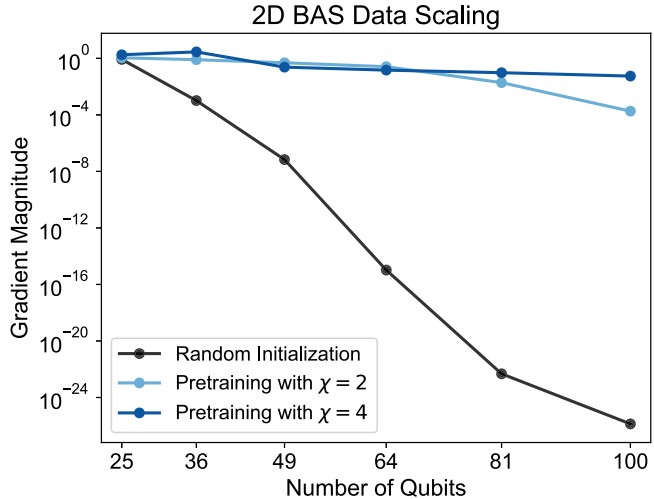

**Fig. 4 | Gradient magnitude scaling for a QCBM with the KL divergence loss function and the BAS dataset.** For the pretrained cases, we train MPS with bond dimensions $\chi = 2$ or $\chi = 4$, decompose them into one layer of SU(4) gates while optimizing the fidelity, and extend that layer to a 2D topology using identity gates. The gradient magnitude, i.e., the 2-norm of the gradient vector, is then evaluated on an MPS-based quantum circuit simulator for practical feasibility. While the gradient magnitude of the randomly initialized circuits decay exponentially with the number of qubits, the pretrained cases exhibit significantly more stable behavior. After $9 \times 9 = 81$ qubits, the gradients for the $\chi = 2$ pretraining start to decay and are surpassed by the $\chi = 4$ case.

required to escape barren plateaus, but our findings suggest that this is in fact not exponentially demanding using a synergistic framework jointly leveraging TNs and PQCs.

The avoidance of barren plateaus, as indicated in Figs. 3 and 4, is vital to ensuring the trainability of PQCs and their viability on quantum hardware. Vanishing gradient variances imply that gradient magnitudes also vanish[19], which leads the estimation of parameter gradients on quantum hardware to require a number of measurements which grows exponentially in the number of qubits. Additionally, barren plateaus have been linked to the phenomena of *cost concentration* and *narrow gorges*[25], which hinder the ability of gradient-based and gradient-free optimizers to find high-quality solutions, as well as the existence of large numbers of low-quality local minima[24], which present further difficulties in learning. Aside from improving the training performance in practice (as seen in Fig. 2), stable gradient variances (such as those in Figs. 3 and 4) hint that a finite (or at worst, non-exponential) number of quantum circuit evaluations may be sufficient to estimate parameter gradients and perform PQC optimization on quantum hardware in a scalable manner.

## Discussion

This work introduces a synergistic training framework for quantum algorithms, which employs classical tensor network simulations towards boosting the performance of PQCs. Our framework allows a problem of interest to be attacked first with the aid of abundant classical resources (e.g. GPUs and TPUs), before being transitioned onto quantum hardware to find a solution with further improved performance. By moving the work of quantum computers to improve on promising classical solutions, rather than finding such solutions de novo, we ensure that scarce quantum resources are allocated where they are most effective, setting up parametrized quantum algorithms for success.

Assessing the performance of our methods on generative modeling and Hamiltonian minimization problems, we found that PQCs initialized with this synergistic training framework not only obtained better training losses using identical quantum resources, but also

exhibited qualitatively improved optimization behavior, with deep quantum circuits transformed from being practically untrainable to reliably converging on high-quality solutions. A study of gradient variances and magnitudes shows the promise of this method for avoiding barren plateaus and related worst-case guarantees, which for randomly initialized PQCs lead to gradients which decay exponentially in both the number of qubits and the depth of the circuit. For PQCs initialized using classically obtained TN solutions however, we observed gradient variances and magnitudes which remain essentially constant with respect to size, even showing a slight increase in deeper circuits. At system sizes up to 100 qubits, we witnessed a change of trends that favored pretraining with larger bond dimensions in order to keep up the favorable scaling. Thus, our results suggest that classical computing resources do not need to drastically increase in order to keep mitigating barren plateaus in PQCs to a very strong degree. These results point towards the promise of this framework for enabling PQCs to scale to large number of qubits, thereby unlocking the latent capabilities of quantum computers for optimization and learning problems which remain out of reach for purely classical methods.

These findings naturally open up several related questions. We have employed MPS as our classical TN ansatz, whose bond dimension $\chi$ determines the classical resources allocated for PQC initialization, but have yet to characterize the performance of our method on problem sizes requiring very large values of $\chi$. While we consistently find larger bond dimensions to yield increased PQC performance in our synergistic framework, we also anticipate an eventual need to employ more sophisticated TN models whose topology is better adapted to the connectivity of the circuit architecture at hand. To this end, initializing PQCs using tree tensor networks is a natural next area of study, as the simple canonical forms available to such models permit a straightforward extension of the decomposition procedure used here[33]. We anticipate the use of more flexible TN models to lead to further improvements in the performance of quantum algorithms, complementary to those identified for the use of larger values of $\chi$.

Despite our evidence suggesting that the influence of barren plateaus can be alleviated with moderate classical resources, there remain significant challenges to overcome before PQCs can find practical application and tackle problems that are currently out of reach for purely classical methods. While further algorithmic improvements are beneficial and perhaps required, the success of sufficiently powerful PQCs with dozens or hundreds of qubits ultimately hinges on the ability to engineer quantum computers with sufficiently low error rates.

As a final remark, we note that our synergistic framework highlights the benefits of moving beyond the adversarial mindset of "classical vs. quantum" which is typical of discussions surrounding so-called "quantum supremacy". By embracing the rich connections between classical TN algorithms and PQCs, we show that good use can be made of the complementary strengths of both. Moving forward, we believe that the existence of powerful classical simulation methods should not be seen as an obstacle on the path to demonstrating practical quantum advantage, but rather as a guide to help quantum methods find their way[12].

## Methods
### Parametrized quantum circuits
Parametrized quantum circuits (PQCs), and in particular the so-called *variational quantum algorithms* (VQAs), are the centerpiece of a family of quantum algorithms that aim to solve practical problems on near-term quantum devices with a limited number of qubits and noisy operations. The parameters $\theta$ of PQCs are usually optimized (or "trained", in the context of learning from data) according to a loss function

$$\mathcal{L}(\theta) = \mathcal{L}(\psi_\theta), \tag{2}$$

where $\psi_\theta$ is the wavefunction of the quantum state prepared by the PQC. Unlike on classical hardware, one does not have explicit access to the prepared state. Therefore, the loss $\mathcal{L}(\theta)$ needs to be estimated using quantum circuit measurements. PQCs are commonly trained via gradient descent methods, such as finite distance gradients or the *parameter shift rule*[47–49], or via gradient-free optimizers such as CMA-ES[42]. For an in-depth introduction to PQCs and VQAs, as well as a broad overview of their potential applications, we refer to ref. 5.

Quantum circuit Born machines (QCBMs)[14] are quantum models for solving generative learning tasks, and without loss of generality, encode probability distributions over binary data as measurement probabilities of a wavefunction prepared by a PQC. The probability assigned to a binary string $\mathbf{x}$ by a QCBM with circuit parameters $\theta$ is given by the Born rule,

$$q_\theta(\mathbf{x}) = |\langle \mathbf{x} | \psi_\theta \rangle|^2, \tag{3}$$

where the parametrized wavefunction $\psi_\theta$ encodes the distribution $q_\theta(\mathbf{x})$. QCBMs are capable of representing complicated probability distributions[50–54], while still permitting a direct means of generating samples from any learned distribution by measuring the associated wavefunction $\psi_\theta$. However, much is still unknown about the performance of QCBMs on near- to mid-term quantum devices, especially when modeling complex real-world datasets[55–57].

Many methods exist for training a QCBM to minimize a problem-specific loss function, which depends on a dataset $\mathcal{D}$ of size $|\mathcal{D}|$ and circuit parameters $\theta$. The loss function we use here is the Kullback-Leibler (KL) divergence between the QCBM distribution $q_\theta$ and the evenly weighted empirical distribution $p_\mathcal{D}$ associated to $\mathcal{D}$, given by

$$\mathcal{L}(\theta) = \mathrm{KL}\left(p_\mathcal{D} || q_\theta\right) = \mathbb{E}_{\mathbf{x} \sim p_\mathcal{D}(\mathbf{x})} \left[ \log \frac{p_\mathcal{D}(\mathbf{x})}{q_\theta(\mathbf{x})} \right]$$
$$= -\log(|\mathcal{D}|) - \frac{1}{|\mathcal{D}|} \sum_{\mathbf{x} \in \mathcal{D}} \log(q_\theta(\mathbf{x})). \tag{4}$$

For non-uniform weighting, the last line of Eq. (4) must be replaced by the appropriate expectation $\mathbb{E}_{\mathbf{x} \sim p_\mathcal{D}(\mathbf{x})}$.

The variational quantum eigensolver (VQE)[58] is a prototypical example of a variational quantum algorithm. The goal in VQE is to find the ground state $\psi_0$ or the ground state energy $E_0$ of a Hamiltonian $H$, which can be found by minimizing the variational energy function

$$\mathcal{L}(\theta) = E(\theta) = \langle \psi_\theta | H | \psi_\theta \rangle \tag{5}$$

of the parametrized trial wavefunction $\psi_\theta$ on a quantum computer. This is done by sampling $\psi_\theta$ in multiple bases to estimate the expectation values of each operator in the Hamiltonian $H$ with finite precision. The VQE algorithm can be used to calculate important properties of Hamiltonians in domains of significant practical interest, for example computing the energy of a molecule in the setting of quantum chemistry. In this setting, the qubit Hamiltonian is obtained from the fermionic Hamiltonian of the participating electrons using, for example, the Jordan-Wigner transformation[59]. Given the practical nature of the problem, and the decades of classical computational techniques towards solving such high-value problems, gave rise to highly specific quantum circuit ansätze and parameter initialization[26,27].

### Tensor networks
Tensor networks (TNs) are linear-algebraic models first developed for representing and classically simulating statistical models and complex many-body quantum systems[60], but they have more recently also been employed as machine learning models[31,61–63]. Tensors are generalizations of vectors and matrices to higher dimensions. The number of axes in a tensor is often called its *order*, where order-1 and order-2 tensors represent vectors and matrices, respectively. A $N$-qubit wave

function is arguably most naturally represented by an order-$N$ tensor where every axis has dimension 2. One approach to reduce the complexity of handling these large tensors with exponentially many entries is to factorize them into a network of lower-order tensors, which, when multiplied together (commonly called *contracted*), recover the original wave function. Depending on the dimension of the axis resulting from the factorization, the tensors can be efficiently stored and used for computation.

The manner in which the tensors are contracted together is determined by an undirected graph, with different graphs determining different TN architectures. The nodes of each such graph correspond to *cores* of the TN, while the edges correspond to *indices* or *links* of the TN, describing tensor contractions to be carried out between pairs of tensor cores along one or more links. For applications to quantum simulation, the number of nodes $N$ in a TN can, for example, be equal to the number of qubits in the quantum computer, and the topology of the $N$-node graph determines the forms of entanglement which can be faithfully reproduced in the classical TN simulation.

In this work, we utilize *matrix product states* (MPS), computationally tractable TN models whose tensors are connected along a line graph (Fig. 1). The tensors are order-3 tensors in the bulk and order-2 tensors at the boundary. Each tensor contains a physical index representing the qubit, and so-called virtual link that connect to neighboring tensor cores. MPS have a long history in the ground state computation of quantum 1D spin chains[64,65] via the *density matrix renormalization group* (DMRG) algorithm[66], as well as for the efficient simulation of quantum computers with limited entanglement[67]. Despite their simplicity, MPS with sufficient bond dimension can simulate any $N$-qubit wavefunction, making them a natural first model for many TN applications. The expressivity of an MPS is determined by its bond dimension $\chi$ (i.e., the dimension of the shared link between neighboring tensors), a quantity associated to the edges of an MPS which sets an upper bound on the amount of entanglement achievable in a simulated quantum state[68]. In cases where the entanglement of a quantum state is greater than an MPS is able to exactly reproduce, the *singular value decomposition* (SVD) may be used to find a low-rank MPS approximation of the state with near-optimal fidelity.

Although we focus on MPS, our results can be straightforwardly extended to more complicated TN architectures, allowing for different tradeoffs between expressivity and classical computational complexity[69].

One application of MPS here is as *tensor network Born machines* (TNBMs)[63], generative models which represent a probability distribution using a simulated quantum wavefunction parametrized by a TN. As such, they form the tensor network analog to QCBMs described in Sec. IV A, and we utilize TNBMs to provide the classical solutions to them.

While QCBMs and TNBMs are similar mathematically, with both model families using the Born rule to parametrize classical probability distributions, they nonetheless have distinct complementary benefits in real-world applications. QCBMs are fully quantum models which are able to leverage advances in quantum hardware to better reproduce the correlations present in complex datasets, but are limited by the state of current noisy intermediate-scale quantum (NISQ) devices. By contrast, TNBMs can take full advantage of recent developments in classical computing hardware, notably the development of powerful graphical/tensor processing units (GPUs/TPUs), but are fundamentally limited in their expressivity by the extent of entanglement they are able to simulate efficiently. Additionally, the analytically explicit construction of TNBMs enables exact calculation of probabilities $q_{\theta}(\mathbf{x})$ (see Eq. (3)) and gradients of a loss function $\mathcal{L}$ with respect to the model parameters $\boldsymbol{\theta}$. The complementary strengths of both models naturally motivate the development of hybrid quantum-classical Born machine models, but this is complicated in practice by the difficulty of converting between these models. Throughout this work, we specifically consider TNBMs implemented by 1d MPS, as opposed to general TN structures that this model allows.

## MPS to PQC mapping

The parameters of PQCs and TNs can in principle be interconverted freely, with the circuit topology of a PQCs itself forming a TN via classical simulation, and with TN canonical forms[68,70] facilitating the representation of a TN as a PQC. In practice though, there are several issues that arise with the latter conversion. The quantum circuits associated with a direct conversion from TNs to PQCs are composed of unitary gates acting on multi-level qudits of varying size, whose compilation into gate sets of real-world quantum computers is itself a non-trivial problem (e.g., see[37]). In the general case of bond dimension of $\chi$, an MPS will be mapped to a quantum circuit containing multi-qubit gates acting on $\lceil \log_2(\chi) \rceil + 1$ qubits per gate. Much preferred however is a decomposition into two-qubit gates. This is practical for a variety of reasons. For instance, many quantum hardware realizations natively implement two-qubit gates, removing computational overhead in applying multi-qubit operations. Additionally, two-qubit gates can be more sparingly parametrized, in contrast to the exponentially increasing number of parameters needed to fully control multi-qubit gates. Despite these challenges, in the following, we find strong evidence that the use of an efficient and high-performance conversion method permits MPS of increasing size and complexity to boost the performance of PQCs within several real-world applications.

We use the MPS decomposition protocol developed in ref. 33, which augments the analytical decomposition method of ref. 37 with intertwined constrained classical optimization steps on the circuit unitaries. Using this protocol, transferring the MPS to a PQC results in $k$ layers of SU(4) unitaries with a next-neighbor topology, also called *linear* or *staircase* layers. We note that this decomposition is performed fully on classical hardware. The choice of an appropriate value for $k$ is a hyperparameter of the decomposition, and the quality of the decomposition for a fixed $k$ is limited by the entanglement present in the MPS. Fortunately, the decomposition protocol used in this work allows for sequential growing of the circuit up to a desired fidelity. We refer to Supplementary Note 4 for a more detailed description of the decomposition protocol used throughout this work.

One may wonder how this process is efficient on classical hardware. This is the case because the created linear quantum circuit layers tend to result in the MPS having a lower bond dimension than before. Generally speaking, if the MPS was computationally feasible beforehand, it should also be feasible to decompose it via this technique. This is opposed to alternative approaches of brute-force optimization of the linear layers. In such cases, the intermediate states reached during optimization are not guaranteed to represent an MPS with $\chi_{max}$ equal to or less than that of the original MPS.

To have a chance at improving the previously found MPS results, one needs to extend the linear layers with additional gates that would have been infeasible to simulate classically, i.e., the bond dimension $\chi$ of the MPS would need to be increased, which is likely not possible at a point where one is planning to continue optimization on a quantum computer. Extending the quantum circuit can either come in the form of increased circuit depth, more flexible entangling topologies, or both. In our work, from the $k$ linear layers, we extend only the final layer of SU(4) gates to an all-to-all topology, that is, a layer containing SU(4) gates between all pairs of qubits. The free parameters of those additional gates are drawn from a normal distribution with zero mean and small standard deviation to not significantly alter the mapped quantum state. Notably, we then train all existing gates in the circuit, and not just the additional gates. We refer to Supplementary Note 4.B for details on the SU(4) gate circuit ansatz as well as the possible decomposition of such gates into Pauli-gates, as well as to Supplementary Note 2 for a brief study of the effect of adding additional gates to the mapped MPS quantum circuits.

## Data availability

The data generated in this study have been deposited in the following GitHub repository: https://github.com/MSRudolph/Synergy-PQC-TN.

## Code availability

The code used to generate the data in this study has been deposited in the following GitHub repository: https://github.com/MSRudolph/Synergy-PQC-TN. Your access to and use of the downloadable code (the "Code") contained in this Section is subject to a non-exclusive, revocable, non-transferable, and limited right to use the Code for the exclusive purpose of undertaking academic, governmental, or not-for-profit research. Use of the Code or any part thereof for commercial or clinical purposes is strictly prohibited in the absence of a Commercial License Agreement from Zapata AI (https://zapata.ai/contact/).

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

## Acknowledgements

The authors would like to acknowledge Vladimir Vargas-Calderón, Brian Dellabetta, Dmitri Iouchtchenko, Peter Johnson, Yudong Cao, Kaitlin Gilli, Dax Enshan Koh, and Mohamed Hibat-Allah for their helpful feedback on early versions of this manuscript. The authors would like to acknowledge the ORQUESTRA® platform by Zapata Computing Inc. that was used for collecting the data presented in this work.

## Author contributions

M.S.R. and A.P.-O. conceived the initial proposal for the synergistic framework. J.M. contributed to the interpretation of the synergistic framework in its final form. M.S.R. wrote the code used in this work and performed most numerical simulations. D.M. performed the numerical simulations for Fig. 4. J.C. provided optimized MPS models for the VQE simulations. J.M., J.C. and A.A. provided relevant expertise in tensor network methods. A.P.-O. helped supervise and coordinate the efforts in this work. All authors regularly analyzed the numerical results and contributed to the final version of the manuscript.

## Competing interests

The authors declare the following competing interests: M.S.R., D.M., and A.P.-O. were employed by Zapata Computing Canada Inc. during the development of this work. J.M., J.C., and A.A. were employed by Zapata Computing Inc. during the development of this work.

## Additional information

**Peer review information** : *Nature Communications* thanks Kerstin Beer, Shi-Ju Ran, Jinguo Liu and the other, anonymous, reviewer(s) for their contribution to the peer review of this work. A peer review file is available.

