## [Peer Review File · Nature Communications]

Synergistic Pretraining of Parametrized Quantum Circuits via Tensor NetworksREVIEWERS' COMMENTS:

Reviewer #1 (Remarks to the Author):

Dear authors, dear editors,

This work presents a procedure for initialisations of parametrised quantum circuits, which avoids barren plateaus and improves the performance and speed of the circuit training. The initialisation is done by pre-optimising on classical hardware using tensor networks. Compared to a traditionally (randomly or nearly-identity) initialised VQA, the pre-optimisation avoids Barren plateaus, which describe the vanishing of the loss gradient and lead to the non-trainability of more extensive quantum circuits.

The authors support their claims by presenting various numerics on three optimisation experiments. The presentation of the numerics is convincing. A link to the complete code would be needed to make the work easy reproducible, which is not given in this work. An often-used method in our field is sharing the code via a GitHub repository.

The main paper is overall well written. Only the introduction to tensor networks is too short or nearly missing. Other than that, I have only a few remarks.

The appendix needs some additional work. I expected to find more details about the methods used and presented results there. However, it mainly only contained some short comments on the optimisation methods. Please find my suggestions and a summary of the paper below.

Overall, this paper convincingly presents that combining classical and quantum algorithms can be helpful. Since the method prevents Barren plateaus, a widespread problem in quantum computing, I am sure the proposal is significant to the field. After working on the introduction to tensor networks, adding more details to the appendix and making the code public, this work can be published in Nature Communications from my point of view.

Best wishes,

Kerstin Beer

--- Suggestions ---

Abstract and title:

Reading the abstract of this work, I expected this paper to explain how the VQA initialisation using TNs is done and which tasks need which initialisation. Further, I expected numerical proofs of comparisons with random initialisation and the here proposed task-specific

initialisation.

After working through the paper, I can say that the wording “task-specific initialisation” is misleading. Instead, I would describe the method by using a classical algorithm to pre-optimize the VQA and optimize it on quantum hardware. The title describes this better.

Introduction:

The introduction is good to read. I have only one remark: Add details on how a concrete realisation of this proposal is missing in Ref. 33 and what you do in this work to fill this gap.

Figure 1:

The middle lines in the MPS on the very left are thicker than the other lines.

The look of this combined graphic would be even more excellent if you remove all the shadows.

In general, I like the idea of the graphic with the laptop and the atom using black and blue colours a lot. I understand that the blue in the picture stands for quantum and black for classical computation. Nevertheless, why is then the only quantum-optimised curve black, not blue? I would change the colours of the plots and use different black/grey shades for the $x=2$, 4 and 8.

II. Methods:

B. Tensor Networks: What does “tensor contraction of low-order tensors” mean? In general, this chapter gives not much information about what TNs are, only how you use them. This needs improvement.

C. MPS to PQC mapping: What do you mean by “all-to-all topologies”? Explain the difference between linear layers here or in Appendix A. Further, you could add the information on whether the “decomposition protocol developed in Ref. [32]” is done on classical or quantum hardware.

III. Result and discussion:

Why do you introduce TNBMs but do not use the terminology in the results sections?

Does the “best performance” (“represents the best performance among 6 independently initialised runs”) mean the lowest value after a fixed number of iterations? This could be more clear.

“increased classical resources (as quantified by the bond dimension χ) leads to improved

performance of the trained quantum circuits”: Is there a bound after which increasing the classical resources does not lead to improvement anymore? If this is not clear yet, make a short remark.

You only mention the two data sets and the Hamiltonian, but not the three training tasks belonging to this kind of data. Explain better what the optimisation goals are.

The last paragraph of this chapter seems not to describe results. Maybe there is another place for this information.

Figure 3:

To clarify what the numbers in the circles mean, add “ $N =$ number of Qubits” under topologies.

Appendix:

Appendix A: I wish that the phrase “decompose the classically trained MPS into k linear layers of two-qubit gates, and extend only the final layer into an all-to-all topology” would be explained in all detail in the appendix. It is also unclear why you do not extend all layers to an all-to-all topology. The appendix would be a good place to give the reader a better understanding, maybe even with nice graphics.

Appendix B: Explain what you mean by “bootstrapped median and 25-75 percentile of the losses” and “2d-correlated dataset, whereas the MPS solutions are 1d-correlated”.

Appendix C: Cite the DMRG method. Why is a single qubit gate $U(3)$, not $U(2)$? The notation is not clear to me. Do you mean a normal distribution (“initialised by sampling a normal distribution N ”) on the parameters of the gates?

Appendix D: The “max”, for example, in the formula, should be text with appropriate distances. Further, I would mention (D1) in the main paper. This is what I was missing while trying to understand your task at first.

Appendix methods: The short introduction to the methods in the main paper (PQCBMs, VQE, TNBMs) was okay, but the appendix needs an additional section where the methods are explained in detail to the reader. Starting with a general introduction to VQAs and TNs would make things perfect.

Appendix experiments: Another appendix explaining the data sets and problems (cardinality dataset, dataset of bars and stripes images and nearest-neighbour spins Hamiltonian) in detail would be highly recommended.

--- Summery ---

The authors of this paper present a procedure for task-specific initialisations of PQCs using tensor networks, which avoids barren plateaus and improves the performance and speed of the circuit training.

Introduction

Quantum devices will be rare. Hence we will focus on quantum technologies, which can solve problems which are difficult for classical algorithms but of high practical interest: Parametrised quantum circuits promise quantum advantage in quantum chemistry, material science and quantum machine learning. However, using generic PQC methods and parameter initialisations, Barren plateaus and problems with local minima are often noticed. However, task-specific initialisations are studied only for some examples in quantum chemistry but not so much in other areas.

The bar for quantum to overcome classical algorithms is raised by classical simulations of PQC based on tensor networks (TN). These simulations can describe PQC efficiently when the intermediate states have limited entanglement. The TN finds a state and converts it to the parameters of a PQC. It is possible to continue the PQC optimisation on quantum hardware. This paper shows that this improves solving different problems with PQCs consisting of up to 20 qubits. Previous work stated the general idea of using TN models for initialising PQCs. Here, the concrete realisation is done. Another work was restricted to the bond dimension 2.

Methods

A. Parametrized Quantum Circuits

PQCs, or in particular VQAs, are quantum circuits whose parameters are optimised by a loss function based on measurements on the wavefunction of the by the circuit prepared state.

1. Quantum Circuit Born Machines

Quantum circuit Born machines (QCBMs) can encode complicated probability distributions over binary data as measurement probabilities of a wavefunction prepared by a PQC. It is still possible to generate samples from the learned distribution by measuring the wave function. For training a QCBM, the Kullback-Leibler divergence, which compares the QCBM distribution and the data set distribution, is used in this work.

2. Variational Quantum Eigensolver

The variational quantum eigensolver (VQE) tries to find a Hamiltonian H 's ground state or energy by minimising the energy function.

B. Tensor Networks

Tensor networks (TNs) are linear-algebraic models which can be used for machine learning or to simulate many-body quantum systems. The ladder can be expressed as a tensor contraction of low-order tensors. These contractions can be expressed in a graph describing the TN architecture. The MPS in this work are expressed in line graph TN models. MPS can simulate any N-qubit wavefunction, but the bond dimension gives an upper bound on the amount of entanglement.

Tensor Network Born Machines

Tensor network Born machines (TNBMs) generate a probability distribution using quantum wavefunction parametrised by a TN. QCBMs and TNBMs have complementary strengths, but converting between these models in practice is complicated.

C. MPS to PQC mapping

The MPS gained by TNs have to be converted to PQCs. It is a non-trivial problem to decompose the gained multi-qubit gate circuits into a circuit of implementable two-qubit gates. The authors propose in this work an efficient and high-performance conversion method. The protocol converts a MPS to a PQC in k layers of $SU(4)$ unitaries with a next-neighbour topology. This technique would also be doable if the MPS was computationally feasible in terms of the bond dimension beforehand. Improving the MPS before turning it into a PQC would include increasing the bond dimension and would likely lead to too much overhead when continuing to optimise on a quantum computer.

III. Results and discussion

The performance of the initialisation method is explored by comparing quantum circuits trained with or without this method. The latter means these are initialised randomly or as gates close to the identity. For the comparison, quantum circuits with linear or all-to-all topologies of fully parametrised $SU(4)$ gates are used. Randomly or identity initialisation of the circuits leads to worse final losses than using the in this work presented method. With increasing classical resources in terms of the bond dimension of the MPS, the final performance of the trained quantum circuit increases. Barren plateaus are entirely avoided.

In the first experiment, the performance of the QCBM is tested with two data sets: the cardinality dataset and the dataset of bars and stripes images. Further, a VQE experiment uses nine qubits with a nearest-neighbour spins Hamiltonian. These experiments use 3 or 4 layers of $SU(4)$ gates initialised in random or near-identity unitaries or with those found by MPS solutions. For the comparison, the same circuit layout and number of parameters are used.

The gradients are measured with respect to the YY entangling gate contribution between

qubits 1 and 2. The not with the method initialised circuits lead to a plateau at much higher KL values. However, in some cases initialising with MPS of bond dimension two is insufficient to improve the optimisation process.

The plots show also that the final MPS losses (dashed lines) are also improving by the quantum hardware. However, the imperfect decomposition of the MPS into a low number of two-qubit gate layers leads to gaps between the loss in the MPS step and the starting loss on the quantum hardware.

To study the trainability of the PQCs, the authors did numerics on the barrenness, the variance of parameter gradients. The results are robust in different choices of the parameter for which the gradients are estimated. The here presented method not only avoids barren plateaus but even can increase the variance with circuit depth. It is essential to avoid barren plateaus since the PQCs can not be trained.

IV. Conclusion

This paper presents a framework to boost the performance of PQCs. TNs are used to find a classical solution which is then improved on quantum hardware. Remarkable is that circuits with a high number of qubits and depth, which are practically untrainable without the method, can find high-quality solutions using the authors' proposal. For example, generative modelling and Hamiltonian minimisation problems are shown. It is an open question how the method works on huge values of the Bond dimension. Probably more flexible TN models are needed. In contrast to the common adversariality between classical and quantum methods, this work shows that we can make use of the complementary strengths of both.

Appendix A: Performance enhancement when extending linear layers

In this work only the final layer is extended to an all-to-all topology. This gives better results also with random initial parameters.

Appendix B: Median Optimization Performance from Sec. III

The authors added statistics on the losses presented in the paper. The synergistic approach with any Bond dimension outperforms the random and identity initialisation.

Appendix C: Implementation Details of the MPS and PQCs

1. MPS optimization

The KL divergence loss is optimised by the density matrix renormalisation group (DMRG) method.

2. SU(4) gates

The two-qubit gates are decomposed into four single-qubit rotations, XX, YY and ZZ (KAK decomposition)

3. PQC simulation

The additional gates in the synergistic framework and all the gates in the near identity initialisations were chosen by sampling a normal distribution. The results on quantum hardware are just simulated.

4. PQC optimization

The PQC optimisation is done in a Python implementation of the CMAES optimiser.

5. Gradients

The gradient was calculated with a finite-distance gradient estimator.

Appendix D: MPS decomposition

The MPS decomposition tries to find a sequence of circuit layers which is, applied on the zero state, close to the target wavefunction. Furthermore, additional circuit layers and optimisation steps can improve precision.

Reviewer #2 (Remarks to the Author):

In this work, the authors propose to combine the classical simulation of tensor network (TN) with quantum circuit models, in order to efficiently deal with the real-world data in the noisy intermediate-scale quantum era. The topic is timely and their proposal is indeed valuable to the community of quantum computing, quantum machine learning, and tensor network. But there are several issues on the manuscript that need to be resolved before being accepted by any journal.

1. The idea and main results about the “synergistic training framework” should be published eventually, but I do not think this work meets the criteria of nature communications. The key point of this paper is to initialize the parameterized quantum circuit (PQC) by classical simulation, which is then further optimized by quantum computing with a modified architecture. This idea is not new, I have to say. The questions we care about are how to initialize and how to modify the architecture. The authors made some meaningful attempts to answer these questions, thus I recommend to accept it (after resolving the issues below) but in a less selective journal.

2. The scalability of the proposed scheme should be discussed. For the Heisenberg model considered in this paper, it just contains 9 qubits, which is far too small. Their simulations

reach at most 20 qubits. Note that in, for instance Ref. [61], the number of qubits is already more than 150. Is there a specific reason for not trying on, say, 100 qubits?

3. A relevant question is how the complexity of the proposed scheme scales with the number of qubits. They mentioned that the final layer is extended to an “all-to-all” topology. Will this layer significantly increase the complexity, and prevent taking a large number of qubits?

4. By “task-specific”, do the authors mean that the corresponding classical TN algorithm for the specific task is applied to obtain the MPS? Please explain this term when it appears for the first time in the main text.

5. The authors mentioned several advantages of their method, but the existing schemes also possess such advantages. Meanwhile, some references are missing and not cited properly.

(a) For transforming MPS to PQC, the authors missed two important references, which are Nat. Commun. 1, 149 (2010) and Phys. Rev. Applied 18, 024013 (2022). The authors mention that “trained MPS with bond dimension $\chi = 2$ were exactly decomposed into a staircase of two-qubit gates” and cite Ref. [34]. This can also be done by the methods proposed in the two references mentioned above and Ref. [61]. Particularly, χ is not restricted to be 2 for Phys. Rev. Applied 18, 024013 (2022).

(b) The authors claim that the circuit in the proposed scheme just contains two-qubit SU(4) gates and the number of layers can be taken flexibly, which is also true for the schemes proposed in Refs. [61] and PRA 104, 042601 (2021).

(c) The authors stated that “In the general case of bond dimension of χ , an MPS will be mapped to a quantum circuit containing multi-qubit gates acting on $\lceil \log_2(\chi) \rceil + 1$ qubits per gate”. This is true for Nat. Commun. 1, 149 (2010), but not true for many references including those mentioned above.

6. In the 10th line of the abstract, I suggest not to claim this proposal as the “first” utilization of TN for identify a quantum state. This idea in general has been discussed for years.

Reviewer #3 (Remarks to the Author):

In this manuscript, the authors introduce a matrix product state (MPS) inspired quantum circuit architecture and its parameter initialization scheme, and apply this new framework in some variational quantum algorithms that proposed for near-term intermediate scale quantum (NISQ) devices. They show clear numeric evidences that the proposed synergistic framework has advantage over both classical tensor network and the same circuit ansatz with randomly initialized parameters.

The idea in the paper is straight-forward. A classical MPS is pretrained, which is then mapped to a parameterized quantum circuit using some known deterministic procedure. To achieve better performance than its classical counter part. Some extra quantum gates

initialized to near identity are inserted to the circuit. The authors claim such a framework can avoid gradient vanishing. I think the absence of gradient vanishing problem in this architecture is not very hard to understand since the matrix product state inspired quantum circuit itself does not have the gradient vanishing problem in some applications (see Ref [65]), the extra near-identity gates should not change the gradient amplitudes too much.

Avoiding the cost concentration or the gradient vanishing problem is the central issue in the field of NISQ variational quantum algorithms. Using a known classical structure to avoid the cost concentration is an inspiring direction to go. The drawback is also clear, it requires a classical MPS structure that can already avoid the cost concentration in that application. While the numeric results are impressive, to meet the high standard set by Nature Communication, I would expect there can be a theoretical understanding that can serve as a guideline for future variational quantum ansatz design. There should also be some proof about the limitation of this method. It is too good to be true if the gradient does not vanish for a general target problem, because then we can solve any computational-hard problem with the proposed framework.

This paper is well written. I have only one minor comment to improve the writing,

* Fig 1. the top and bottom color schemes for quantum gates are inconsistent.

Response to Reviewer #1

We thank Reviewer #1 for their exceptionally detailed and rigorous review. All of the Reviewer's comments have strong merit and have helped us to improve the quality and inclusiveness of our work.

Overall, Reviewer #1 finds our numerical results convincing, which show that informed parameter initializations derived from optimized tensor networks strongly mitigate the effect of barren plateaus on the trainability of parametrized quantum circuits. The Reviewer believes that such synergistic frameworks combining classical and quantum algorithms are significant for the wider community and recommends this work for publication in Nature Communications after addressing their remarks. However, before addressing the Reviewer's remarks, we feel compelled to acknowledge that a full open-sourcing of the framework's source code is not possible since it is part of the proprietary software suite of Orquestra™ and Zapata Computing Inc. However, as a company we are committed to the reproducibility and availability of our results, so we have added a note in the data availability section stating that the necessary code and results will be available upon direct request from anyone interested in using it for verification purposes.

Our responses to Reviewer #1's detailed comments can be found below, with changes highlighted in green in the marked-up PDF version of the revised manuscript. Before that however, we wanted to briefly summarize some additions to our revised manuscript, which were inspired by the feedback of all Reviewers. First, we would like to draw attention to the new Figure 4, which demonstrates a large-scale scalability study of gradient magnitudes, with and without pretraining, up to 100 qubits. These numerical results definitively show that classical computing resources do not need to increase drastically (or even exponentially) to keep up the favorable effect of mitigating barren plateaus for parametrized quantum circuits. This is a result which, to our knowledge, has not been provided by any other work. Furthermore, we added a new Appendix A, which zooms into individual models, with and without pretraining, and depicts representative gradient magnitude distributions. In accordance with all other results provided in our manuscript, this figure shows that synergistically pre-trained models systematically break the trend that randomly initialized models show in terms of their gradient distribution. These two additions to our manuscript increase our confidence that such TN-PQC frameworks have the potential to address large fractions of trainability issues caused by barren plateaus and to bring us collectively closer to leveraging near-term quantum devices to their fullest potential.

The introduction is good to read. I have only one remark: Add details on how a concrete realisation of this proposal is missing in Ref. 33 and what you do in this work to fill this gap.

We thank Reviewer #1 for their comment on the readability of our work. Based on this and other Reviewers' feedback, we have decided to add a Related Work section, which highlights important prior art and contextualizes our work in this light. We hope that this choice satisfies some of the Reviewer's requests.

After working through the paper, I can say that the wording “task-specific initialisation” is misleading. Instead, I would describe the method by using a classical algorithm to pre-optimize the VQA and optimize it on quantum hardware. The title describes this better.

While we believe the wording “task-specific initialization” reflects an important aspect of our method, based on the Reviewers’ feedback, we now see that it may not be the right choice of words. We have thus replaced almost all instances of this phrase with more nuanced descriptions.

The middle lines in the MPS on the very left are thicker than the other lines.

This is actually intentional, and is a common shorthand in the TN literature, where thicker lines indicate larger entanglement quantified by higher bond dimension. We have increased the width of those lines even more to highlight that this is not a mistake.

The look of this combined graphic would be even more excellent if you remove all the shadows. In general, I like the idea of the graphic with the laptop and the atom using black and blue colours a lot. I understand that the blue in the picture stands for quantum and black for classical computation. Nevertheless, why is then the only quantum-optimised curve black, not blue? I would change the colours of the plots and use different black/grey shades for the $x=2, 4$ and 8 .

We thank Reviewer #1 for this feedback. Almost all of these suggestions have been included in the revised version of Figure 1. We did, however, refrain from changing the only quantum-optimized curve from black to blue because we used gray/black-tone colors throughout our work to indicate random initializations. The color contrast should therefore be seen as “pre-trained PQC” (blue) vs. “the rest” (gray/black). Reviewer #1’s comment on the curves on the classical side thus hit the nail on the head.

Tensor Networks: What does “tensor contraction of low-order tensors” mean? In general, this chapter gives not much information about what TNs are, only how you use them. This needs improvement.

Based on Reviewer #1’s feedback, we have realized this weakness in our work and have attempted to make the section more inclusive to a general audience that may have less experience with the TN-specific jargon. We now start with a general introduction of tensors and tensor network representations as graphs and move more slowly towards MPS, which plays a primary role in our work. We hope that the revised section is more in tune with what Reviewer #1 expected from a brief introduction to tensor networks.

MPS to PQC mapping: What do you mean by “all-to-all topologies”? Explain the difference between linear layers here or in Appendix A. Further, you could add the information on whether the “decomposition protocol developed in Ref. [32]” is done on classical or quantum hardware.

We have clarified both of these points in the revised manuscript.

Why do you introduce TNBMs but do not use the terminology in the results sections?

We thank Reviewer #1 for pointing out this lack of consistency. We now use TNBM for all occasions where we are referring to MPS that are used to classically implement Born machines.

Does the “best performance” (“represents the best performance among 6 independently initialised runs”) mean the lowest value after a fixed number of iterations? This could be more clear.

This is now more clearly specified in the Results section.

“increased classical resources (as quantified by the bond dimension χ) leads to improved performance of the trained quantum circuits”: Is there a bound after which increasing the classical resources does not lead to improvement anymore? If this is not clear yet, make a short remark.

Reviewer #1 has clearly identified one of the future directions of work, which is to explore how much pretraining is ideal, not sufficient, or too much in such a synergistic framework. In our new Figure 4, we highlight a scenario where stronger pretraining can be beneficial, and our discussion surrounding Figure 7 in Appendix C and also in the main text showcases that there can certainly be pathological cases where pretraining using MPS is detrimental to the following PQC optimization. We have tried to better highlight these aspects of our results throughout the main text.

You only mention the two data sets and the Hamiltonian, but not the three training tasks belonging to this kind of data. Explain better what the optimisation goals are.

We have added a brief description of the goals of the generative modeling and ground-state search tasks. We thank the Reviewer for bringing to our attention this opportunity for clarifying our manuscript.

The last paragraph of this chapter seems not to describe results. Maybe there is another place for this information.

We understand Reviewer #1’s reservations, as we also had difficulty with deciding the proper placement of this paragraph. The context in this paragraph is vital for appreciating the full impact of our empirical alleviation of barren plateaus, and thus should appear after the results. While we considered moving this paragraph to the Conclusions section, we felt it distracted too much from the broader outlook presented there, hence the current placement.

To clarify what the numbers in the circles mean, add “ N = number of Qubits” under topologies.

We thank the Reviewer for this feedback. The Figure has been adjusted accordingly.

Appendix A: I wish that the phrase “decompose the classically trained MPS into k linear layers of two-qubit gates, and extend only the final layer into an all-to-all topology” would be explained in all detail in the appendix. It is also unclear why you do not extend all layers to an all-to-all

topology. The appendix would be a good place to give the reader a better understanding, maybe even with nice graphics.

We understand the sentiment that echoes through various of the Reviewers' comments. Not only have we attempted to add some detail helpful to this particular Appendix, but throughout the manuscript, we have now highlighted more that the best way to extend the gates in a circuit is likely highly dependent on both the problem and the topology of the quantum hardware being utilized.

Appendix B: Explain what you mean by “bootstrapped median and 25-75 percentile of the losses” and “2d-correlated dataset, whereas the MPS solutions are 1d-correlated”.

We now elaborate on the bootstrapping technique used for enhancing the robustness of uncertainty estimates and elaborate on what is meant by the bias mismatch between the data and the MPS. The latter point is further discussed in the main text when the datasets are introduced and in the discussion surrounding the new Figure 4.

Appendix C: Cite the DMRG method. Why is a single qubit gate $U(3)$, not $U(2)$? The notation is not clear to me. Do you mean a normal distribution (“initialised by sampling a normal distribution N ”) on the parameters of the gates?

We thank the Reviewer for highlighting that the DMRG method should also be cited in the Appendix, and for identifying this typo ($U(3)$ has been corrected to $U(2)$, corresponding to fully general single-qubit rotations). Additionally, we added more clarity in the main text to our descriptions of how the additional parameters are initialized.

Appendix D: The “max”, for example, in the formula, should be text with appropriate distances. Further, I would mention (D1) in the main paper. This is what I was missing while trying to understand your task at first.

The “max” subscript of the bond dimension is now properly in text form. Furthermore, we now directly mention in the main text that more practical details about $SU(4)$ gates can be found in the Appendix.

Appendix methods: The short introduction to the methods in the main paper (PQCBMs, VQE, TNBMs) was okay, but the appendix needs an additional section where the methods are explained in detail to the reader. Starting with a general introduction to VQAs and TNs would make things perfect.

We agree with the Reviewer that our introduction to PQC/VQAs and TNs was quite brief in the original manuscript. Given that our work has the greatest relevance to the PQC/VQA community, we have focused on Reviewer #1's request to write a longer stand-alone introduction to TNs (the former topic being more familiar to readers). We hope that these changes make our manuscript more understandable to the broader quantum computing community.

Appendix experiments: Another appendix explaining the data sets and problems (cardinality dataset, dataset of bars and stripes images and nearest-neighbour spins Hamiltonian) in detail would be highly recommended.

Following Reviewer #1's comments, we have added more details to the main text, which we hope gives sufficient detail on these points.

Response to Reviewer #2

We thank Reviewer #2 for the exceptionally valuable context they provided to our work. Reviewer #2's comments were appreciative of the practical impact of our synergistic TN-PQC framework, and how this overcomes many of the worrying trainability barriers that PQCs face due to barren plateaus. At the same time, Reviewer #2 identified an important limitation of our results in reaching the high bar for publication in Nature Communications, namely the need to extend our numerical evidence to larger system sizes. Reviewer #2 also raised several specific concerns about properly contextualizing our contributions within the existing literature.

We are glad to report that Reviewer #2's comments have been a great help in improving our manuscript, which is significantly revised from its original version. While we have made many changes, we would especially like to highlight to Reviewer #2 the new Figure 4 and the surrounding discussion, which demonstrates the effect of MPS pretraining on system sizes up to 100 qubits (a substantial enhancement from our previous results and conclusions based on system sizes up to 20 qubits). These new results give compelling evidence that the dramatic improvements offered by our method in mitigating barren plateaus and enhancing the trainability of PQCs persist into large system sizes, and don't require a dramatic increase of classical computing resources. We consider this to be great news for the quantum computing community, and mention that these trainability benefits are not mirrored to the same extent in any other work that we are aware of.

In addressing Reviewer #2's comments about contextualizing our results (see below), we found it helpful to add a new standalone Related Work section. We hope this better highlights the relationship of our current work to many of the relevant past results that have shaped the community's thoughts about combining quantum circuits and tensor networks inside hybrid frameworks.

This is not to say that these are the only improvements. We have accumulated all reviews to provide more helpful details and explanations throughout our main manuscript and the Appendix. Of note is the new Appendix A which attempts to further characterize the gradient magnitudes of PQCs that were initialized by MPS solutions. **We find that these pre-trained models exhibit qualitatively changed character in terms of their gradient distribution**, which is consistent with all of our other results, and with the new Figure 4 in particular.

We duly hope that Reviewer #2 views these improvements as significant enough to consider recommending the publication of this revised manuscript. Next, we address and expand on each of the comments and concerns raised by the Reviewer.

1. The idea and main results about the “synergistic training framework” should be published eventually, but I do not think this work meets the criteria of nature communications. The key point of this paper is to initialize the parameterized quantum circuit (PQC) by classical simulation, which is then further optimized by quantum computing with a modified architecture. This idea is not new, I have to say. The questions we care about are how to initialize and how to modify the

architecture. The authors made some meaningful attempts to answer these questions, thus I recommend to accept it (after resolving the issues below) but in a less selective journal.

We thank the Reviewer for their positive comments on the contributions of our work. They pinpointed exactly how we understand our work— as an actual practical realization of this TN-PQC framework that has been on experts’ minds for years. As we show through our detailed numerical studies, including a drastically improved scalability analysis (following Reviewer #2’s comments), the consequences of pretraining quantum circuits with tensor networks are not as straightforward as one might believe. We now know that pretraining with moderate classical resources can keep up the very favorable scaling in terms of gradient variance and magnitudes, whereas one may have thought that a drastic or even exponential increase in classical computing would have been necessary. A similar result apparently holds for pretraining PQCs with 1D TNs, even though the target tasks exhibit correlations that do not follow a 1D topology. While we show that naive circuit expansions can fall short to significantly enhancing the TN solution, in general, it is still vastly superior to random initializations. This is particularly striking in our new Figure 4, where we illustrate this in systems sizes up to 100 qubits.

On the other hand, we do believe that our work reflects a different focus than most other publications on this topic. As we aim to highlight in the manuscript, the ultimate goal of this framework is not to pre-train quantum circuits, but rather to solve challenging real-world problems. High-performance decomposition algorithms open up the possibility of attaining the maximum classical performance and only then, if strictly necessary, move the workload to a quantum computer. Our work then covers the study of the following PQC performance in the context of the many worst-case trainability guarantees that the literature reports. We believe that our work thus represents a significant step forward in the development of practical applications of PQCs, which we show only requires moderate classical guidance to be set up for success. The improved evidence we present to support this is not in small part due to the excellent and challenging suggestions of Reviewer #2, which we now address in greater detail.

2. The scalability of the proposed scheme should be discussed. For the Heisenberg model considered in this paper, it just contains 9 qubits, which is far too small. Their simulations reach at most 20 qubits. Note that in, for instance Ref. [61], the number of qubits is already more than 150. Is there a specific reason for not trying on, say, 100 qubits?

This comment by Reviewer #2 sparked the biggest update to our manuscript, namely the new Figure 4, which examines the gradient magnitudes of PQCs extended to a 2D topology containing up to 100 qubits. For this, we utilize an MPS-based simulator with automatic model differentiation, which allows us to calculate exact gradients in a scalable manner. As we describe in detail below, the focus of [2] (Ref. [61] of the original manuscript) was completely different from our work, but the decomposition and gradient computation methods we use are similarly scalable, and thus present no inherent limitations to scaling up our gradient magnitude characterizations to yet larger system sizes.

We should point out that scaling up the full optimization results of Figs. 1 and 2, where the PQC topology is extended and trained using quantum resources, remains challenging beyond 20 qubits. This

limitation is by design, as the extension of the classically-initialized circuit is designed to leverage entanglement that is only available on quantum computers. Nevertheless, our new Fig. 4 clearly demonstrates that a reasonable amount of classical resources can provide significantly improved PQC trainability by mitigating the issue of barren plateaus, and that increasing these classical resources continues to provide value at large system sizes.

3. A relevant question is how the complexity of the proposed scheme scales with the number of qubits. They mentioned that the final layer is extended to an “all-to-all” topology. Will this layer significantly increase the complexity, and prevent taking a large number of qubits?

Reviewer #2 is correct in that extending from linear layers to all-to-all topologies presents difficulties, and the use of this layer was for practical convenience in working with arbitrary numbers of qubits at smaller problem sizes. The new Fig. 4 presents a more realistic choice of circuit expansion which is problem-inspired, namely extending the circuit to a 2D topology for a problem that exhibits strong 2D correlations. While the question of the best circuit extension will ultimately depend on both the target problem and the quantum hardware being utilized, we emphasize that this extension *should* be one that is difficult to simulate classically. While our classical initialization method, which is applied before the circuit extension, must be classically scalable, the circuit that is ultimately optimized on a quantum device should make use of the full entanglement achievable by the quantum device in order to attain any sort of non-classical advantage.

4. By “task-specific”, do the authors mean that the corresponding classical TN algorithm for the specific task is applied to obtain the MPS? Please explain this term when it appears for the first time in the main text.

Reviewer #2 highlights a phrasing that was also commented on by Reviewer #1, and we have consequently replaced almost all instances of this phrase with more nuanced wording.

5. The authors mentioned several advantages of their method, but the existing schemes also possess such advantages. Meanwhile, some references are missing and not cited properly. For transforming MPS to PQC, the authors missed two important references, which are Nat. Commun. 1, 149 (2010) and Phys. Rev. Applied 18, 024013 (2022).

We thank Reviewer #2 for bringing these missing references to our attention, which we have made sure to include in our revised manuscript. More broadly, we were inspired by Reviewer #2’s comments to add a new Related Work section, which we hope allows readers to more easily compare our work with important prior techniques.

We agree with many of the specific points raised by Reviewer #2 (addressed in detail below) but should first point out that these comments are entirely focused on the MPS to PQC decomposition method we employ, which was the focus of [1]. The focus of this paper is rather on using large-scale classical computing resources to enhance the performance of variational quantum algorithms and is in no way bound to any specific MPS to PQC decomposition method. Indeed, any efficient decomposition method satisfying the following decomposition criteria (DC) can be freely used within our synergistic framework

to boost the quality of PQC solutions in a scalable manner: (DC1) It must accept as input an MPS state of any bond dimension χ , (DC2) It must output a circuit of two-qubit gates of any desired depth, and (DC3) It must converge to the original MPS state vector as the circuit depth increases. We realize this point was somewhat implicit in our original manuscript, and so have added a clear statement of this fact in the Related Work section of our revised manuscript.

In light of this, the points brought up by Reviewer #2 actually lend more evidence to the value of our synergistic approach since the use of other MPS to PQC decomposition methods (such as the ones described by Reviewer #2) may actually improve on the already-respectable results we found.

5a. The authors mention that “trained MPS with bond dimension $\chi = 2$ were exactly decomposed into a staircase of two-qubit gates”... This can also be done by the methods proposed in the two references mentioned above and Ref. [61]. Particularly, χ is not restricted to be 2 for Phys. Rev. Applied 18, 024013 (2022).

We thank Reviewer #2 again for sharing these references, which are promising decomposition methods. Unfortunately, for our purposes, [2] (Ref. [61] of the original manuscript) does not satisfy criteria (DC3), as seen in the plateauing fidelity with circuit depth reported in [1] and [2]. [3] (Phys. Rev. Applied 18, 024013 (2022)) does not satisfy (DC2), since it is restricted to producing one layer of d -qubit gates, where d is generally greater than 2.

5b. The authors claim that the circuit in the proposed scheme just contains two-qubit $SU(4)$ gates and the number of layers can be taken flexibly, which is also true for the schemes proposed in Refs. [61] and PRA 104, 042601 (2021).

As described above, [2] does not suit our needs, but [4] (PRA 104, 042601 (2021)) very much does. We became aware of this important reference after the preparation of our original manuscript and have actually started using a generalization of this decomposition method in a more recent extension to our synergistic method (manuscript in preparation). We are in complete agreement with Reviewer #2’s positive assessment of this method and its potential for boosting the performance of our synergistic framework yet further.

5c. The authors stated that “In the general case of bond dimension of χ , an MPS will be mapped to a quantum circuit containing multi-qubit gates acting on $\lceil \log_2(\chi) \rceil + 1$ qubits per gate”. This is true for Nat. Commun. 1, 149 (2010), but not true for many references including those mentioned above.

While an important prior work, [5] (Nat. Commun. 1, 149 (2010)) does not output two-qubit gates, violating (DC2), and is therefore not a good fit for our synergistic method.

6. In the 10th line of the abstract, I suggest not to claim this proposal as the “first” utilization of TN for identify a quantum state. This idea in general has been discussed for years.

We think this comment might arise from a misreading of our abstract, which states that our method “first utilizes tensor network (TN) simulations to identify a promising quantum state, which is then converted into gate parameters of a PQC”. This simply describes our synergistic optimization procedure, where one first finds an approximate solution using TNs, and then transfers it over onto a quantum circuit. We do not make any claims about being the first to utilize TNs for identifying quantum states, which is indeed a well-established idea. We are open to changing this particular sentence in a future revision in order to avoid confusion, but we currently feel that it is a fair description of the framework.

References

- [1] Manuel S Rudolph, Jing Chen, Jacob Miller, Atithi Acharya, and Alejandro Perdomo-Ortiz. “Decomposition of matrix product states into shallow quantum circuits,” arXiv:2209.00595 (2022).
- [2] Shi-Ju Ran. “Encoding of matrix product states into quantum circuits of one-and two-qubit gates,” *Physical Review A* (2020).
- [3] Prithvi Gundlapalli and Junyi Lee. "Deterministic and entanglement-efficient preparation of amplitude-encoded quantum registers." *Physical Review Applied* (2022).
- [4] Peng-Fei Zhou, Rui Hong, and Shi-Ju Ran. "Automatically differentiable quantum circuit for many-qubit state preparation." *Physical Review A* (2021).
- [5] Marcus Cramer et al. "Efficient quantum state tomography." *Nature Communications* (2010).

Response to Reviewer #3

We thank Reviewer #3 for their kind remarks on our “impressive” numerical results and their assertion that the synergistic framework we propose is an “inspiring direction”, which shows an “advantage over both classical tensor network and the same circuit ansatz with randomly initialized parameters.”

The concerns of Reviewer #3 mostly revolved around the reasons for the effectiveness of our method, both in relation to vanishing gradient results in MPS models, as well as the need for a better analytic understanding of how our method works. We provide some clarification below on this former point, describing how the success of our method is independent of vanishing gradient results for classical MPS. We address the latter point first, though, where we present new results that shed light on how our method avoids vanishing gradients and identifies potential pitfalls along the way.

We assure Reviewer #3 that our results are not, in fact, “too good to be true” and thank them for bringing up the aforementioned concerns, which have led to significant improvements in our manuscript (see the new Appendix A for one example). We hope the results described below provide the sort of nuanced understanding Reviewer #3 had been desiring, which more clearly highlights the novelty and value of our synergistic method. Furthermore, our new Fig. 4, depicting a study of gradient magnitudes up to 100 qubits, makes us confident that our results are not a fluke. All of our numerical studies point in the same direction, which is that a moderate amount of classical computing (and in particular not exponential) is sufficient to qualitatively transform untrainable PQC’s into ones that can reliably improve on the classical solution.

I would expect there can be a theoretical understanding that can serve as a guideline for future variational quantum ansatz design. There should also be some proof about the limitation of this method.

We thank Reviewer #3 for this challenge, which has inspired much lively discussion among the authors. Our first attempt was to attempt a rigorous proof of the success of our method in avoiding vanishing gradients in certain situations. However, after several false starts (rigorous results of this nature typically merit a stand-alone publication and are very difficult to obtain), we identified another approach that proved very insightful. Rather than looking at aggregate gradient variance, we decided to examine detailed histograms of the magnitudes of individual gradients within circuits that had been classically initialized using our method.

What we found, which is the focus of our new Appendix A, is quite revealing. The distribution of gradients in a randomly initialized circuit is very regular, falling off exponentially with increasing magnitude (as expected by vanishing gradient results). By contrast, pre-trained circuit gradients are much more broadly distributed in magnitude and don’t seem to follow a clear statistical pattern. While the pre-trained parameters exhibit gradients that are larger than randomly initialized ones, many of the large gradients occur within the newly added gates, which represent the quantum contributions to the state space of the variational model. In this sense, the effectiveness of our method shouldn’t be seen as a

purely-classical or purely-quantum phenomenon, but rather from the use of a classically-derived initialization to enhance the behavior of the quantum model within this larger state space.

The primary limitation of our method can be best understood through the notion of “inductive bias”, as described in Sec. III of our manuscript. When we use increasingly performant classical models to initialize a quantum circuit, we are biasing the model to continue making use of the correlations representable by that model (between nearest neighbors on a line-shaped topology), and in some cases this can lead the model to not taking full advantage of the new quantum gates. We saw this, for example, in some of the random seeds used in the 2D bars and stripes experiment (Fig. 2 and Fig. 7, middle panels), where using an MPS with a larger bond dimension in some cases delivered worse performance under the quantum training than by using an MPS with smaller bond dimension. Still, the results are vastly superior to random initializations. As our method is leveraged within more challenging problems, we expect to encounter new successes and new limitations for our method and anticipate the well-established tools from classical machine learning (such as regularization and hyperparameter tuning) to be invaluable in navigating these tradeoffs.

I think the absence of gradient vanishing problem in this architecture is not very hard to understand since the matrix product state inspired quantum circuit itself does not have the gradient vanishing problem in some applications (see Ref [65]), the extra near-identity gates should not change the gradient amplitudes too much.

Reviewer #3 is correct that [1] (Ref [65] of our original manuscript) uses numerical simulations to identify a reduction in vanishing gradients using their Q-MPS architecture. However, we should point out first that the model they use differs significantly from ours, including that (a) Q-MPS is a different circuit topology from ours (compare our Fig. 1 to their Fig. 2), and (b) Q-MPS uses quantum number conserving blocks (we do not), which are claimed to be the primary source of their observed non-vanishing gradients.

More importantly, though, we should mention that randomly initialized MPS *have*, in fact, been rigorously proven to exhibit barren plateaus in many settings, as described in [2] and [3]. We use a global loss function in our work, which [2] suggests would lead to vanishing gradients if using a randomly initialized MPS circuit. The fact that we don’t observe this problem here is further evidence of the effectiveness of our synergistic approach beyond what was already described in our manuscript.

Using a known classical structure to avoid the cost concentration is an inspiring direction to go. The drawback is also clear, it requires a classical MPS structure that can already avoid the cost concentration in that application.

As proven in [2] (and described in detail above), simply using MPS-structured circuits does not on its own avoid the problem of cost concentration. This can be witnessed in Fig. 3, where gradient variances decay exponentially with linear layers (closely related to MPS in terms of their entanglement structure) as the circuit depth is increased. The success of our synergistic approach should therefore be looked for elsewhere, in particular in our novel hybrid optimization method. Consequently, the only real limitation of our approach is the requirement that we use classical tensor networks, which considering the flexibility and proven effectiveness of these models, we feel is hardly a drawback at all.

After concluding this point-by-point response, we hope the substantially revised version, encompassing the additional clarifications highlighting the value and novelty of our synergistic method, and supported by the new significantly larger simulations and insights, are aligned with the desired revisions required from the Reviewer for our article to be considered for publication in Nature Communications.

References

- [1] Jin-Guo Liu, Yi-Hong Zhang, Yuan Wan, and Lei Wang. "Variational quantum eigensolver with fewer qubits." *Physical Review Research* (2019).
- [2] Zidu Liu, Li-Wei Yu, L-M. Duan, and Dong-Ling Deng. "Presence and Absence of Barren Plateaus in Tensor-Network Based Machine Learning." *Physical Review Letters* (2022).
- [3] Enrique Cervero Martín, Kirill Plekhanov, and Michael Lubasch. "Barren plateaus in quantum tensor network optimization." *Quantum* (2023).

REVIEWERS' COMMENTS

Reviewer #2 (Remarks to the Author):

I appreciate the great efforts of the authors on improving the manuscript. I feel all my comments have been well responded. Particularly, the data shown in Fig. 4 are quite persuasive, suggesting that the barren plateaus can be well avoided by choosing a proper initialization using the TN methods with classical simulation. The barren plateaus bring an essential challenge for optimizing quantum circuits. The evidence on avoiding the barren plateaus should be given with the simulations of large numbers of qubits, which is the case of the newly added Fig. 4. Therefore, I agree to accept the manuscript in its current form.

Reviewer #4 (Remarks to the Author):

In the present manuscript, the authors consider the problem of trainability for parametrized quantum circuits (PQC), which are often considered promising candidates for attaining a practical quantum advantage in near-term quantum devices. However, one of the known problems is that optimization landscapes typically have barren plateaus when random initial states are used, preventing effective trainability. In this work, the authors propose a method to remedy this problem by utilizing classically pre-optimized initial states. The optimization relies on efficient and scalable tensor-network methods. In particular, the authors claim that using such initial states improves the trainability of the PQC by leading to better quality solutions while avoiding barren plateaus, as compared to generic initializations.

My overall assessment is that the idea presented here has merit, the numerical evidence is rather convincing, and the presentation of the manuscript is coherent. However, I find that the present work is not original enough to deserve publication in the highly selective Nature Communications. I now elaborate on that.

The main novelty of the present manuscript is to combine a controlled and efficient quantum circuit decomposition for MPS, together with the idea of using pre-optimized initializations to PQC. My criticism is that the decomposition algorithm was already introduced and analyzed in Ref.[31] (itself inspired by Ref. [37]), while the idea of using pre-

optimized states is hardly original. I have carefully read section II in the manuscript, including the criteria the authors state for a suitable decomposition method, and I am not aware of any other work in the literature with a similar context that these criteria are met simultaneously. However, very similar ideas of using pre-optimized initial PQC have appeared before (some are mentioned in the main text and also by Reviewer #2) and, although this work is a step forward, I do not find it novel enough for the present journal.

In terms of result, I find the most important contribution to be the evidence provided for the absence of barren plateaus when the proposed initial states are utilized. Although I find this to be an advancement in the context of PQC, I am not convinced about its practical implications. The authors suggest in several places in the manuscript (including the subtitle "Short-cutting the Race to Practical Quantum Advantage") that their findings are likely of practical importance for near-term devices. However, it seems to me that this ignores the crucial role of noise in quantum devices. Even if the problem of barren plateaus due to random initialization is solved, it has been suggested that noise has a distinct role in inducing (qualitatively distinct) barren plateaus (Ref.[22] in the manuscript). Moreover, the noise seems to severely limit the practical applicability of variational algorithms [see, e.g., Nat. Phys. 17, 1221–1227 (2021)], making me skeptical about the actual impact of the findings of the present manuscript.

To conclude, I was asked to comment on whether the authors have satisfactorily addressed the issues raised by Reviewer #3.

-- Regarding the first question on theoretical understanding: This is a natural question to ask, but the present work focuses on numerical experiments and it is fine that the authors cannot fully resolve this, which would possibly deserve a separate publication.

-- Regarding the second question on the connection between a vanishing gradient and the architecture: I find the answer of the authors convincing, i.e., MPS alone is not enough to avoid barren plateaus.

Reply to Reviewer #2

We are grateful to Reviewer #2 for their helpful comments on our manuscript throughout the revision process, and are glad that they are appreciative of our significant revisions. We thank the Reviewer for their recommendation to publish our work in Nature Communications.

Reply to Reviewer #4

We thank Reviewer #4 for their positive feedback on our work. In particular, we are glad that the Reviewer finds our numerical evidence on mitigating the effects of barren plateaus important and convincing. In the following, we will reply to the points raised by the Reviewer.

The main novelty of the present manuscript is to combine a controlled and efficient quantum circuit decomposition for MPS, together with the idea of using pre-optimized initializations to PQC. My criticism is that the decomposition algorithm was already introduced and analyzed in Ref.[31] (itself inspired by Ref. [37]), while the idea of using pre-optimized states is hardly original. I have carefully read section II in the manuscript, including the criteria the authors state for a suitable decomposition method, and I am not aware of any other work in the literature with a similar context that these criteria are met simultaneously. However, very similar ideas of using pre-optimized initial PQC have appeared before (some are mentioned in the main text and also by Reviewer #2) and, although this work is a step forward, I do not find it novel enough for the present journal.

While Reviewer #4 describes the main novelty of our work as the introduction of a framework for pre-training PQCs using classical means, we feel that our contributions extend beyond this. Although previous work has anticipated that some form of pre-training using tensor networks should yield improvements in the performance of PQCs, our work provides much clearer insights into the magnitude of these improvements. We find it striking that overcoming barren plateau issues may only require modest and very attainable amounts of classical resources. These results appear to hold even with fairly generic circuit ansätze, and with system sizes of at least up to 100 qubits. To our knowledge, this constitutes the clearest numerical evidence of this key insight in the context of quantum-classical frameworks.

In terms of result, I find the most important contribution to be the evidence provided for the absence of barren plateaus when the proposed initial states are utilized. Although I find this to be an advancement in the context of PQC, I am not convinced about its practical implications. The authors suggest in several places in the manuscript (including the subtitle "Short-cutting the Race to Practical Quantum Advantage") that their findings are likely of practical importance for near-term devices. However, it seems to me that this ignores the crucial role of noise in quantum devices. Even if the problem of barren plateaus due to random initialization is solved, it has been suggested that noise has a distinct role in inducing (qualitatively distinct) barren plateaus (Ref.[22] in the manuscript). Moreover, the noise seems to severely limit the practical

applicability of variational algorithms [see, e.g., Nat. Phys. 17, 1221–1227 (2021)], making me skeptical about the actual impact of the findings of the present manuscript.

We thank Reviewer #4 for raising this important point, and have decided to add a new paragraph to the Discussion section making this context regarding noise-induced barren plateaus available to readers. While it is true that our work does not solve all the challenges facing PQCs before they can find practical applicability, we nonetheless believe our framework provides important tools for also addressing these remaining challenges. Firstly, we highlight a specific role where quantum computers can be impactful with less overall resources, namely as an extension of classical methods. Furthermore, the independence of our framework from any specific decomposition method means that we can use a decomposition algorithm that anticipates the impact of noise on a given PQC ansatz when choosing our pre-trained initializations, likely aiding in the mitigation of noise-induced barren plateaus. Irrespective of when this may happen, we believe that our subtitle “Short-cutting the Race to Practical Quantum Advantages” remains a fair assessment of the broader implications of our work. However, we understand that our current title may imply inflated expectations about the state of PQCs. The revised title of our manuscript, “Synergistic pretraining of parametrized quantum circuits via tensor networks”, in our opinion, addresses this worry.

To conclude, I was asked to comment on whether the authors have satisfactorily addressed the issues raised by Reviewer #3.

-- Regarding the first question on theoretical understanding: This is a natural question to ask, but the present work focuses on numerical experiments and it is fine that the authors cannot fully resolve this, which would possibly deserve a separate publication.

-- Regarding the second question on the connection between a vanishing gradient and the architecture: I find the answer of the authors convincing, i.e., MPS alone is not enough to avoid barren plateaus.

We are glad that Reviewer #4 finds our additions adequately addressing the concerns of Reviewer #3. We agree it would clearly be invaluable to have a quantitative theory on the mitigation of barren plateaus via pre-training, and we are optimistic that our present work might inspire such research in the future.